# Introducing the Extended Safety Fractal: Reusing the Concept of Safety Management Systems to Organize Resilient Organizations

**DOI:** 10.3390/ijerph17155478

**Published:** 2020-07-29

**Authors:** Bart Accou, Genserik Reniers

**Affiliations:** Safety and Security Science, Delft University of Technology, 2600 Delft, The Netherlands; G.L.L.M.E.Reniers@tudelft.nl

**Keywords:** safety management system, resilience, safety fractal, safety leadership, safety culture

## Abstract

Although mandatory in most high-risk industries, the safety management system (SMS) is often criticized as burdensome and complex. Through its requirement to formalize all main activities, the SMS is perceived as bureaucratic and a vehicle for pure compliance and Safety I (one). Furthermore, the SMS is often detached from an organization’s core activities, goes against local practice and does not deliver the safe performance that was hoped for. By comparing the model behind SMS with specific requirements for process capability, this paper identifies a safety fractal that reflects the basic requirements that are needed to control safety related activities at all levels within an organization. It is further argued that the constituent elements of this safety fractal are particularly suitable to organize resilient performance, provided that resilience is explicitly identified as the safety strategy to follow and, as such, consequently implemented. This approach is then positioned against common safety management concepts as management system maturity, leadership and safety culture, leading to a systematic and a more comprehensive view on how to measure safety performance and resilience.

## 1. Introduction

The concept of a safety management system (SMS), to continuously improve safety of operations, was introduced already some decades ago. Historically, the introduction of SMS can be situated in the context of a changing interest of regulators in the 1970s and 1980s from detailed technical concerns to issues of decision-making. A series of investigation reports following serious accidents (e.g., Herald of Free Enterprise and Clapham Junction) acted as a catalyst, indicating failings of management as a contributing factor [1]. In particular, the Cullen report on the Piper Alpha disaster [2] required, as one of 106 detailed recommendations, to have a safety case accepted by the UK safety authorities for anyone operating an offshore installation. This launched the idea of an auditable SMS that quickly found an audience in high-risk industries like transport (aviation, railways, etc.), the production of dangerous goods, occupational health, etc. This transition from an often very prescriptive safety approach towards an approach that is evidence-driven and based on goal-oriented legislation did not only become normative but even legally mandatory in these industries. Consequently, the establishment and maintenance of an SMS to control all risks related to a company’s operational activities became the basis for certification and regulation [3,4,5,6,7,8].

Maurino [9] describes the introduction of SMS as the logical and coordinated integration of three different tracks that has guided safety and organizational thinking since the 1950s. The first track is system safety engineering which aimed to design for minimum risk [10]. Human factors, the second track, proposed a multi-disciplinary approach to optimize the relationship between people and their operational environment. Business management, finally, introduced the concept of quality management systems and a striving for continuous improvement. In a wider context, when discussing SMS as the formal aspect of the organizational focus surrounding safety, several authors [11,12,13] refer to it as the third age of safety science. This third age not only follows, but also integrates the elements of the first age, focusing on technology, and of the second age, focusing on human behavior and competence.

The introduction of SMS can also be seen as part of a regulatory strategy to place the responsibility for managing safety at the level of the organization best able to do so [6,14]. Rather than blindly complying with prescriptive rules and regulations, they are challenged to identify, in a structured way, what activities are critical for safety and what kind of safety management best fits their particular situation, in order to achieve acceptable levels of safety performance, [7,15,16,17]. In addition, the organizational changes and the rational thinking about safety required for the successful implementation of an SMS are believed to have a positive impact on safety culture [18,19], as well as on an organization’s financial, economic and competitive performance [9,20,21,22].

On the other hand, as reported upon by several authors, success is not guaranteed when implementing an SMS and there are multiple difficulties to overcome. The SMS can be made too burdensome and complex, resulting in processes that are incompatible with an organization’s core activities [9] and going against the common sense found in local practice [8,18]. In addition, Lin [23] reports on a practical gap she identified in companies that tried to incorporate theoretical management models as a tool for resolving issues related to human performance or technical failure. Furthermore, through its requirement to establish procedures and documentation for all main activities, the SMS is often perceived as too normative and bureaucratic [8,18,24,25], pushing companies directly to a “work as imagined” and compliance-focused perspective [19,26]. A similar struggle to shift from an enforcement-based or “control and command” relationship towards a partnership aimed at achieving an agreed safety performance is identified at the level of regulatory bodies [9,24]. This has even led Kelly to conclude that “most regulatory bodies are unable to assess the effectiveness of a company’s SMS” [17], a bold statement that probably finds its origin in the joint finding of various authors that many existing (SMS audit) tools are not linked to defined and established underlying SMS models [27] and do not enable the consistent measurement of the process’ effectiveness to deliver safe performance [18,28,29].

Despite the often well-founded criticism they formulate about its implementation and current use, most of the above-mentioned authors do not argue in favor of abandoning the concept of SMS. On the contrary, they believe that SMS has the potential to better integrate proactive safety management [17] and to make them more resilient by shifting the focus towards more interactive methods to predict and detect undesired outcomes [30]. In that context, also measuring the effectiveness of an SMS or its processes is seen as a way to better capture the true safety state of an evolving organization [1,18,28].

This belief is shared by the authors of this paper, whose research is aiming to revitalize the concept of SMS to deal with the recent paradigm shift in safety management. Hereby, a specific focus is put on linking the generic management activities, aiming to identify and control risks in a systematic way, with the operational activities of the organization that create these risks in the first place. Building on the apparent need for similar feedback loops or plan-do-check-act (PDCA) cycles at the different hierarchical levels in an organization [5,23,31], the second section of this paper explains how the safety fractal is developed. This safety fractal is representing a generic set of requirements that can assure the design of adequate resources and controls for the proper functioning of processes and safety-related activities at all levels in an organization and wider socio-technical system. This is done by comparing the basic principles of process capability [32] with the general requirements of SMS, using the “Dutch Safety Management Model” that has a pedigree that goes back to the first modelling of SMS at the Delft University of Technology in the early 1990s [23]. The third section of this paper further reflects on how this safety fractal can be used to introduce the management of performance variability into SMS and how the basic processes of an SMS can also foster resilience engineering. This is then positioned against common safety management concepts, such as management system maturity, leadership and safety culture, in order to propose improved ways for measuring SMS performance, to organize resilience as well as for in-depth analysis of accidents and incidents, by using the (extended) safety fractal. The use of these models, to address the above-mentioned weaknesses of SMS, finally, is discussed in Section Four.

## 2. Building the Safety Fractal

Despite (or maybe because of) the industry-wide introduction of SMS as the cornerstone for safety management in high risk industries, there is little consensus about what an SMS is and how it should be managed [5,8,25,30]. As a first example, the International Civil Aviation Organization (ICAO) [33], in its Annex 19, defines SMS as “a systematic approach to managing safety, including the necessary organizational structures, accountabilities, policies and procedures”. The same Annex 19 [33] further identifies “safety policies and objectives”, “safety risk management”, “safety assurance” and “safety promotion” as the basic building blocks of an SMS. In a top-down manner, this develops the general safety policy into a number of generic management activities [23]. Similar lists of building blocks or basic elements for an SMS exist across the different literatures and industries. Grote [5] summarized these elements into the following set, representing a common denominator: (1) safety policy; (2) safety resources and responsibilities; (3) risk identification and mitigation; (4) human factor based system design; (5) safety training; (6) safety performance monitoring; (7) incident reporting and investigation; (8) auditing; (9) continuous improvement; (10) management of change.

The European Railway Safety Directive [34,35], on the other hand, defines SMS as “the organization, arrangements and procedures established by an infrastructure manager or a railway undertaking to ensure the safe management of its operations”, highlighting herewith the need to integrate safety in the daily operations of an organization in a formalized way. Unfortunately, this emphasis is not further developed into the published list of “basic elements” for an SMS that shows remarkable similarities with the top-down list of Grote [5].

The quest to make the ideas behind the SMS requirements operational is partly what has been driving this research. On the on the hand, organizations invest in the standardization of management activities, as promoted by SMS principles. On the other hand, it is common practice to elaborate safety rules or procedures at operator level, in order to prescribe how to carry out safety critical tasks or activities. Both are part of a broad historical evolution of safety [36,37] that was denominated “Safety Proceduralisation” by Bourier and Bieder [38]. In addition, most SMS standards or models identify the management of procedures as one of the principal elements, requiring written documentation [39]. Veweire [40] goes even further, by identifying a clear process orientation and a thorough understanding of the company’s operational processes as key differentiating characteristics that set operationally excellent firms apart from other companies.

Proceduralisation of operations at the sharp end can (and most probably should) be very different from proceduralisation of safety management, even within a same organization [36]. Nevertheless, the apparent need for similar feedback loops or cycles at the different hierarchical levels in an organization [5,23,31] gives enough foundation to believe that it is possible to identify a generic set of requirements that can assure the design of adequate resources and controls for the proper functioning of processes and safety related activities at all these levels. This idea of scale invariance, which also features, for instance, the functional resonance analysis method (FRAM) [41] and that is reminiscent of the characteristics of a fractal, explains the name that was given to the model. A fractal is a natural phenomenon or a mathematical set that exhibits a repeating pattern that displays at every scale. It is also known as expanding symmetry or evolving symmetry. If the replication is exactly the same at every scale, it is called a self-similar pattern [42].The following sections will elaborate the consecutive steps that were followed to build this “Safety Fractal”.

### 2.1. Modelling Safety Management Processes 

To be able to comply more effectively with the legal requirements for having an SMS, several industries have attempted an explicit modelling of not only the operational risk-scenarios, but also the safety management processes driving them [23]. With the goal to identify self-similar attributes for safety management, as discussed above, several of these models and standards have been reviewed.

From all reviewed models, the Dutch Safety Management Model appeared to be the most attractive to be used as a starting point for detailed comparison with requirements for process capability. Since the early 1990s the Delft University of Technology started modelling SMS, and the resulting model has a long development history through several consecutive projects. It has been tested and peer reviewed a number of times and has finally been validated through comparison with aviation models (e.g., accident models, technical models, and safety audit programs) and international standards [23]. Furthermore, the model combines management control and monitoring loops, as defined in top-down SMS descriptions, with the systematic logic to represent entities and their activities imposed by the use of SADT (structured analysis and design technique) [1]. This creates a natural bridge to process management, and as a consequence of this approach, the developed generic model already tries to link the traditional SMS elements with the direct aspects of the execution of an action.

The “Dutch” model, which will be used for further reflection and development, contains a nine-step generic management process that is applicable to all delivery systems, managing human and technical performance. In this model, depicted in Figure 1, a closed loop learning system is formed, making a clear distinction is between the internal processes of the management level (i.e., all steps except (4) and (5)) and the output of each delivery at the operational level (i.e., steps (4) and (5)).

The seven steps at the management level define the elements any delivery system should contain to ensure proper safety management:(1)Specify: defining the processes, identifying risks and control measures, based on task analysis of human and technological performance as risk control measure.(2)Provide: ensuring that the measure is designed, built, procured, installed and adjusted to its operating circumstances.(3)Promulgate and train: informing and training workforce to perform the designated actions.(6)Monitor/evaluation: detecting (potential) deviations from the specified functioning of risk control measures and evaluating performance at both the operational and management level.(7)Maintain/change: restoring or improving the functioning of risk control measures, including organizational learning.(8)Collect state of the art: learning from internal and/or external sources to improve operational and managerial performance.(9)Assess risks of proposed changes: identifying the potential impact when deciding to change risk control measures or delivery systems.

Safety management is herewith identified as ”the process to provide the resources and controls designed to ensure that the internal processes are working properly, taking into account the threats that interfere with it”. This focus on the management of processes is modelled at the operational level, by identifying the following steps:(4)Threats and the internal process: the threats that could put in danger the adequate performance of human behavior, the technical system and/or the interaction between both, and that are to be managed by the steps at the management level of the delivery system.(5)Actions executed at the sharp end: the direct execution of the primary work process, by humans and technology, to be compared with the expected performance.

In this model, the identified threats or influencing factors for human performance are either internal (physiological and psychological factors which might influence human information processing, e.g., technical and interpersonal skills, physical fitness, psychological fitness and motivation to commit to safety) or external (factors external to the human which might influence the information processing, e.g., clear and relevant guides, technology-man-machine interface, data and information and environment in which the action needs to be performed). For the technical performance, errors in the design and manufacturing process and in the maintenance and inspection program are identified as main threats. In addition, the possibility of iterating the human threats deeper into another level of human error and organization, to be able to model design and/or maintenance error, is identified. This goes along with the aim of this paper to identify a generic set of self-similar requirements for safety management, applicable at all levels in an organization or wider socio-technical system.

### 2.2. Taking into Account Process Capability

As a next step in the research, this “Dutch” model for SMS was compared with the ISO/IEC 15504 standards on process capability, in order to define a set of requirements that are equally valid at process as well as system level. ISO/IEC 15504 [32] is in fact a set of standards, originally focusing on software development processes and derived from the process lifecycle standard ISO/IEC 12207. The standards contain a reference model defining a process dimension and a capability dimension and therewith provide a structured approach for the assessment of processes.

The standard identifies two principal contexts for the use of process assessment [32]. Firstly, within a process improvement context, process assessment provides the means of characterizing the current practice within an organizational unit in terms of the capability of the selected processes. Analysis of the results identifies strengths, weaknesses and risks inherent in the processes. These provide the drivers for prioritizing improvements to processes. Process capability determination, on the other hand, is concerned with analyzing of the proposed capability of selected processes against a target process capability profile in order to identify some of the risks involved in undertaking a project using the selected processes. For the purpose of building a set of generic requirements that is equally valid at process as well as at system level, primarily the capability dimension of this standard is of use. This capability dimension, as summarized in Table 1, provides a scale of six levels using the following nine process attributes (PA to measure the capability of processes.

Each of these attributes is further detailed with a set of specific requirements. When ordering these detailed requirements of the attributes at process level, according to the steps of the “Dutch” generic safety management process, three (instead of the initial two) levels or types of safety-related activities become apparent:A level of process performance that is modelling the direct functioning of the components that interact during process execution (“doing things”). This is also the level where variation against process specifications can be observed. The direct execution, as well as the sustainable performance, of a process results from the adequateness and capability of the delivery system that is composed of the following levels of process implementation and process control. This is in line with the definition of safety management, introduced by Lin [23], as “the process to provide the resources and controls designed to ensure that the internal processes are working properly to analyse and deal with the threats that interfere with it, so that they are managed to an acceptable level”.A level of process implementation, providing the resources and means to ensure the correct functioning (“doing things right” according to Zwetsloot, [24]) of the process components during process execution. Where the “Dutch” model identifies a behavioral and a technological component as relevant to be managed for control over process execution, the performed analysis identifies the additional need to also manage an organizational component of process design and deployment. This finding emphasizes what already in 1965 was stated by Leavitt, in his diamond: that the performance of people and technology are affected by an organization’s structure and the processes or tasks to be performed (in Deschoolmeester and Braet, [43]). Managing process performance will therefore require not only managing the separate components (an organization’s structure, the tasks or processes to be performed, the people and its technology) but also the possible interactions between them.A level of process control, ensuring the sustainable control of risks related to all activities of the organization in a possibly changing context (“doing the right things” according to Zwetsloot, [24]). The PDCA-cycle that is characteristic for the management system standards and that is also part of the “Dutch” safety management model finds its roots in system thinking and cybernetics [31]. Within a specified system, a process is executed to achieve a defined purpose, according to a specified pattern of progress. Because of external, environmental factors, progress can show some variation that is monitored at regular or irregular intervals to identify the gap between the desired and the actual state of the system. When a gap is identified, the system state is adapted. The identification of variables that allow the measurement of process performance is therefore an essential prerequisite. A similar logic is clearly present in the process attributes of the ISO 15504 capability model [32], resulting in the specification of three elements that together need to ensure the continuous and sustainable control of risks within a changing context: specify, verify and adapt.

In conclusion, comparing relevant models for both SMS and process capability shows a high degree of similarity, enabling the set of generic and scale invariant requirements for good safety management at all levels in an organization we were looking for to be built. The detailed requirements for each identified level that result from this comparison can be found in Table A1 in Appendix A.

### 2.3. The resulting Safety Fractal

Compared to the ‘Dutch” generic SMS model, three distinct process=related type of activities are defined (process performance, process implementation and process control) and an additional organizational component is added to cover the impact that task design and organizational structure could have on process performance. This results in the representation of a safety fractal that is describing, in a self-similar way, a five-step safety management delivery system with the following logic (see also [44]):(1)Specify: the scope and desired outcome of an activity is specified, roles and responsibilities identified, disrupting events are anticipated and risk control measures (rules, barriers) are designed (i.e., work as imagined).(2)Implement—train, equip, organize: all is done to have activities performed by enough competent people, adequate technical resources are put available and maintained, work products and resources to be used are identified and work is planned in detail.(3)Perform: the activity is executed, responding to real life constraints and disturbances (i.e., work as done).(4)Verify: the system’s performance is monitored, i.e., verifying the match between work as designed and work as actually performed, as well as the elements that could affect this performance in the near term.(5)Adapt: it is known what has happened and lessons are learned from experience and the adequate changes to control, or implementation elements, are introduced.

The Figure 2 below graphically represents the found attributes of the safety fractal in the form of a triangle, grouping the composing attributes along the three sides according to the nature of their goal. The left-hand side represents the level of process performance (“doing things”). The bottom side groups the elements of process implementation, providing the resources and means to ensure the correct functioning (“doing things right”) of the process components during process execution. The right-hand side of the triangle, finally, stands for the level of process control, ensuring the sustainable control of risks related to all activities of the organization in a potentially changing context (“doing the right things”). The arrows, in turn, indicate the logical order in which these safety management activities are normally performed.

Together, the implementing and controlling stages define the formal as well as the informal side of safety management and have a direct influence on performance. This representation of the core attributes for safely managing an activity is aligned with the common approach for modelling systems, subsystems and their interactions, as proposed by Wahlström and Rollenhagen [45]. They propose using a control metaphor for the design and assessment of SMS in combination with the concepts of man, technology and organizational and information systems (MTOI) to ensure the continued safety of the operated systems. Wahlström and Rollenhagen [45] further elaborate how this control metaphor, that initially focuses on the safe management of sharp end activities, can also be used for controlling the MTOI systems, as well as different safety management activities, separately and together. A similar line of thought can also be found with other authors [23,46,47,48]. This strengthens our belief that this simple safety fractal model can be applied for all types of activities, including those that form the control and implementing part of it, at every level of aggregation and at every level within a socio-technical system. This line of thought and how best to exploit it will be further developed and explored in the following sections. 

## 3. Organizing Resilience through the Safety Fractal

Mainly for historical reasons, SMS is considered by some authors [49] as a vehicle for reactive safety management. The idea that the concept can also be used to introduce resilience into an organization is, however, gaining ground [19,30]. This view is confirmed by Pariès et al. [25], who states that several safety strategies can fit within an SMS framework, describing the SMS as defining the “piping” of the system, generating safety. This pipework is presented in contrast to the safety strategy, i.e., the models or theories that can help us make sense of the diversity that can be observed in the real world, as the substance that “should flow through the pipes”. Like other authors before us [25,30,50], we propose resilience as the strategy to be used to improve safety in complex systems. In the following sections, the fitness of the safety fractal to host the constituent elements for achieving resilience is first tested. Next, these findings are positioned against common safety management concepts, such as management system maturity, leadership and safety culture, leading to an extended safety fractal that summarizes, in a structured and coherent way, all elements that are required for organizing sustainable and safe performance. 

### 3.1. From Managing Threats to Managing Performance Variability

Many of the cited authors [23,43] still consider the identification and elimination or control of threats and the possible negative consequences that come with it as the objective of safety management. Hollnagel [51,52,53], however, argues that there will always be variability in human performance, individually or collectively. The best option for managing safety is, therefore, not to eliminate this performance variability but rather to monitor the system’s performance so that potentially uncontrollable variability can be caught early on. In addition, creating those conditions that make work succeed should then dampen the critical variability and generate “resilient performance”.

Resilience is defined as the intrinsic ability of a system to adjust its functioning prior to, during, or following changes and disturbances, so that it can sustain required operations under both expected and unexpected conditions [54]. For this research, the safety fractal model, that was developed by looking for synergies between the basic principles of process capability and the general requirements of SMS, was compared with the four potentials that are proposed by Hollnagel [52,53] as necessary for resilient performance (i.e., the potential (1) to respond, (2) to monitor, (3) to learn and (4) to anticipate). This already gives a clear indication that the attributes of the safety fractal only need to be applied with the right mindset, i.e., making the switch from managing threats to managing performance variability, in order to have the potential to generate resilient performance. 

Denyer [55] translates this into four core processes: (1) insight (i.e., interpret and respond to your present conditions), (2) oversight (i.e., monitor and review what has happened and assess changes), (3) hindsight (i.e., learn the right lessons from your experience) and (4) foresight (i.e., anticipate, predict and prepare for the future). He presents these processes as complementary and to be combined with the classical PDCA cycle to offer a structured framework towards continual improvement and innovation in ways that add real value to stakeholders and mitigate the impact of disruptions. He also introduces the concept of “paradoxical thinking” as a solution to manage the inherent tensions between the distinct perceptions that have dominated the discussions on safety management over the past decades. On the one hand, there is the tension between behaviors that are defensive (i.e., stopping bad things happen) and those that are progressive (making good things happen). On the other hand, there is also the tension between behaviors that are consistent and those that are flexible. Leaders then need to demonstrate paradoxical thinking to balance these different approaches and perceptions, in order to find the right fit for their organization. This is building on the reflections of authors like Hollnagel [52], who emphasize the need to complement safety by design (analytical safety) with safety by management (operational safety). Or, even earlier, Wildavsky (1988, ref. in [25]), who described anticipation (predict and prevent danger before damage is done) and resilience (cope with and learn from unanticipated hazards after they have become manifest) as complementary and necessary safety strategies. All these elements mainly point towards the “control” level in the safety fractal (i.e., specify, verify and adapt), confirming the view that safety management is, in essence, controlling the organizational functions and practices that together produce safe performance [47,52].

Logically, the “specify” element of the safety fractal will cover the ability to anticipate and prepare for normal operations and foreseeable hazards. However, several authors (e.g., Le Coze [56]) also argue that part of the response to sudden disruptions should be included in the design of the system. Positioning the seven resilience principles identified by Johnsen [50] in the safety fractal, confirms this view. He argues for graceful and controlled degradation (1), redundancy (2), in having alternate ways to perform a function, flexibility (3) and the reduction of both complexity (4) and coupling (5), all to be integrated during the specification phase of the system based on common mental models (6). 

Johnsen [50] further identifies the management of margins (7) as a key aspect of resilience. This is to ensure that performance boundaries are not crossed through the monitoring of both the slow erosion of the system [30,57], as well as the more dynamic decisions that balance productivity versus safety [58]. This will require continuous monitoring of the system that fits well within the “verify” element of the safety fractal. It is an already long accepted fact that the traditional metrics for measuring safety, which are primarily based on negative outcomes, cannot capture the true safety state of an evolving system [28,49,59,60]. The search for adequate proactive or leading indicators remains, however, a challenge. One of the reasons for this might be that in complex systems, with non-linear interactions, it becomes more difficult to understand the “mechanisms” that lead to risks [52]. Herrera et al. [49] suggest a solution that combines more traditional lagging indicators with indicators reflecting risk influencing factors found through the analysis of incidents and indicators relevant for (variability in) normal operations. In addition, Lofquist [30,61] points to the interactive phase of his integrated safety management model as the most critical phase where discrepancies and undesired outcomes must be dealt with. To be able to notice and act upon safety- and performance-related issues, it should be clear that this will require a transfer of system control from the past towards the present of the process. A move that Pariès et al. [25] also link with a transfer of control from the top towards the bottom of the organization, which will require proper education, training, understanding and experience of the people taking real-time decisions [61]. In addition, Johnsen [50] highlights the need to integrate resilience in the implementation phase of systems, namely through technical solutions, organizational routines and the knowledge and ability of the users of the system, herewith only confirming the logic of the safety fractal. The overview above shows that resilience, as a safety strategy, can be fully embedded by the elements of the safety fractal. On the other hand, it also indicates which elements must be specifically developed and how, in order to create the processes that can optimally support resilient performance.

### 3.2. The Informal Aspect of Managing Safety

Resilience relies heavily on the ability of people, be it at the sharp end or at the blunt end, to adapt what they do in order to ensure that the system continues to function and achieve the set objectives under changing condition [25,54,61]. Denyer [55] for instance, when describing preventive control, mindful action, performance optimization and adaptive innovation as the four perspectives of organizational resilience, relies heavily on an informal and behavioral component. It seems therefore only logical that the contribution of safety culture to the achievement of sustainable and safe performance is also considered.

Unlike the SMS, that is providing the formal foundation for safety management by defining and prescribing what is required through control and implementation processes and arrangements, safety culture is not something that can be agreed between management and workers or between a safety authority and a regulated company based on a norm or standard. Culture should rather be seen as “emerging where people interact and have to accomplish something together” [11,12]. To understand if and how (safety) culture can contribute to improve sustainable safety performance, at least a basic understanding of the complex social processes that create culture within an organization is needed. This will also help to identify the type of internal and/or external interventions by which safety culture might be influenced.

In particular the model developed by Guldenmund [12] on how culture develops within an organization is helpful in the context of this paper. Guldenmund explains that a member of management or staff in an organization, when experiencing a specific situation, develops his or her own perceptions and tries to “make sense”. This becomes his or her individual understanding of reality, influenced in the first place by his or her own individual context (knowledge, individual attitudes, skills and ability, personal characteristics, emotions, state of mind, history, etc.) on which he or she decides or acts. Through an ”interaction” phase, members of the group then exchange meanings, giving rise to mutual adjustments, agreements and expectations with regard to each other’s behaviors, eventually resulting in partly shared understandings. Next, the organization starts officializing a specific set of shared representations and actions, mainly through the “formalization” of structure. This mainly happens through the distribution of tasks, roles and responsibilities and the description of procedures and rules, as well as through more physical structures like technology. These organizational structures, rules and procedures are then instructed and “disseminated” in various forms of communication and education. Through the following step of ‘re-enforcement’, using various organizational processes and with an important role played by leaders, meanings, standards and expectations are finally accepted as the “way to do things”. Members of the group will now share a comparable understanding of reality and structures, which will be the reference for individuals within this group to understand and cope with reality and which will influence the way they make sense of and act on the new situations they experience. These shared patterns of thinking and acting are then what embodies culture.

Through this cycle of (1) sense-making, (2) interacting; (3) formalizing, (4) disseminating, (5) re-enforcing, it becomes obvious that the SMS, which is the instrument par excellence for formalizing safety management, is an important enabler for the development of organizational culture. This leads to the possibility of deploying the SMS as an instrument to exert a positive influence on an organization’s safety culture. To achieve this, the SMS should be built consciously to impact the physical environment, as well as the behavior of employees in a manner that promotes and facilitates sustainable and safe performance. Another aspect that is highlighted by the above cultural development model is that culture, as common patterns of behavior and thinking, is constructed through the interactions between actors. This explains the high importance that is given in most safety culture norms and models to traits like transparency, trust and leadership. The signs given by management through organizational decisions and managerial behavior (listening attitude, recognition, sanctioning, etc.) not only impact the behavior of sharp end operators through positive and negative reinforcement [62], but also impact an organization’s capacity to learn [63]. What is important however, is to understand that this process of cultural development will take place in all situations where a group of people is trying to achieve a common goal. This means that, as earlier identified for the SMS, only a consistent translation and implementation of a consciously chosen and fitting safety strategy into the complete cycle of “cultural enablers” will lead towards an organizational context in which, in due course, a culture can develop that will support sustainable and safe performance.

### 3.3. Developing the Extended Safety Fractal

Based on the above reflections, organizing for resilient, sustainable and safe performance requires a conscious decision for resilience as a safety strategy, with clear choices that provide guidance and direction. In order to have this implemented and “lived” by the whole organization, this strategy then needs to be consistently integrated throughout the organization in all interacting, disseminating and re-enforcing actions, as well as in the more formal SMS. However, strategy implementation is difficult, because it forces people to change their behavior and it requires strong leadership capabilities [40]. Safety leadership can thus be understood as the ability of a manager or staff member to influence behavior so that it becomes safer [64]. The “Leadership in Safety Working Group” [64] identified the following seven general leadership principles that summarize good practice and action principles aimed at safety leaders at all levels in high-risk organizations: (1)create a safety vision that is coherent with the values and principles of management;(2)give safety its rightful place in the organization and management and oversee it on a daily basis;(3)share the safety vision: influence, persuade and promote the flow of information through the hierarchy;(4)be credible: provide a coherent example;(5)promote team spirit and horizontal cooperation;(6)be available on-site to observe, listen and communicate effectively;(7)acknowledge good practice and apply fair sanctions.

When we interpret these general leadership principles within the context developed earlier in this paper, we can only conclude that safety leadership is the combination of having (developed) a safety vision and consequently implementing it through the more informal cultural enablers (i.e., interacting, dissemination and re-enforcing) within an organization or wider socio-technical system.

Management system maturity models, on the other hand, appear to measure the level of implementation of a pro-active safety strategy into the SMS. Maturity models, also including the ISO 15504 model [32] presented above, find their origin in the total quality management movement from the late 1970s. They build on the idea that small, evolutionary steps, rather than revolutionary ones, are the basis for continuous improvement [65]. These quality maturity models mainly provide guidance on how to improve people’s competence, processes and technology (i.e., the formal side of safety management) within an organization in order to move towards sustainable performance. The different stages of maturity should then guide an organization to move from ad hoc, reactive and often chaotic management towards well-established and predictable processes, with continuous improvement as a major objective. These ideas were then picked up to describe maturity levels also for safety management and safety management systems, with, for instance, Zwetsloot [24] describing an occupational, health and safety management system that distinguishes four stages of maturity: (1) ad hoc, (2) systematization, (3) systems approach and (4) proactive and integrative.

To summarize, these elements of SMS and SMS maturity, safety vision, safety culture and safety leadership, that all have been identified as essential to organize resilient and sustainable performance, can be graphically represented in an “Extended Safety Fractal”, as depicted in Figure 3.

The left side of this figure shows at the bottom the original safety fractal, developed in Section 2 of this paper, which represents the effort required for formal and organized safety management. As explained in Section 3.1, sustainable safety management requires the systematic implementation of a safety strategy that is the top of this extended safety fractal. Management system maturity then measures the extent to which the safety strategy is also effectively embedded in the SMS or the underlying processes. The right side of the figure represents the extent to which leaders across the organization promote and support that same agreed safety strategy in their daily activities to achieve sustainable safety management. The bottom side, finally, collects the elements identified in Section 3.2 as the enablers for the development of an organizational culture. The “shared patterns of acting and thinking”, in the middle of the figure, is then the (safety) culture that emerges as a result of the interaction between the surrounding elements. 

## 4. Discussion

The way in which the safety fractal and its extended version are constructed guarantees their applicability at different levels within an organization and even a broader socio-technical system. The possibility of repetition and the self-similarity of the models allows us to intuitively bring the logic of generic SMS processes closer to the operational activities of an organization and to integrate them in a natural way. As argued, the importance of a clear safety strategy for this cannot be underestimated. Resilience as safety strategy, and a focus on managing performance variability rather than eliminating threats, can easily be integrated in the developed models. As a result, both models lend themselves perfectly as a vehicle for the recent paradigm shift in safety management, making optimal use of the experience gained with SMS over the past decades. 

Discussing the resilience of an organization is, however, only of academic value if it is not possible to assess, in advance of accidents and disasters, whether an organization has the required qualities [66]. Measuring the effectiveness of an organization’s processes is seen as a better way to capture the true state of evolving organization and the formal part of safety management, the SMS, looks like the most appropriate starting point. Several authors [17,29,67] have reviewed existing methods and proposed alternative categories for classifying SMS audit and evaluation techniques, with Peltonen [27] even providing an overview of more than 50 existing techniques. 

The most straightforward classification is, however, provided by Cambon [18], who identifies three traditional ways of measuring SMS performance: (1) the analysis of achieved result, (2) the analysis of the approach that is used for setting up the SMS and (3) the comparison of an SMS with an existing reference. The same author also identifies three fundamental attributes of an SMS that need to be captured in order to be able to build a picture of SMS performance: (1) the degree of SMS formalization, which is similar to attributes of the earlier discussed process capability methods, (2) the quality of the implementation, by assessing whether the SMS effectively manages to prevent accidents from happening and (3) the level of ownership of the SMS by members of the staff in the organization, looking at the more informal aspect. The proposed approach then combines a questionnaire, based on the TRIPOD methodology [27], to capture SMS ownership with more traditional audit techniques, like document review, interviews and observations, to result in an integrated picture of SMS performance.

Lofquist [30], on the other hand, when making a case for building resilience into SMS, proposes two areas that can improve the measurement of effectiveness in SMS in order to capture the signals of drift [57] towards and beyond the system’s safe boundaries: (1) the functioning of reporting systems and (2) (improved) safety climate surveys, an idea that also Johnsen [50] promotes. A similar conclusion is made by Hale et al. [66], who state that SMS audit tools can provide the hooks to assess resilience, provided that a closer coupling can be made between the traditional SMS structure and safety culture. This is in line with the description of Hollnagel [10] of resilience as an expression of how people, alone or together, cope with everyday situations by adjusting their performance to the actual operating conditions. With the resilience assessment grid (RAG), he suggests measuring resilience through the proxy of the four resilience potentials (i.e., the potential to respond, to monitor, to learn and to anticipate).

None of these proposals, however, cover the entire extended safety fractal (Figure 3) that summarizes all essential elements needed for an organization to come to a sustainable, safe and resilient performance and which is believed to be the necessary scope for measuring the effectiveness of the safety management of a complex system. This will require a combination of different techniques with clear and explicit focus on an organization’s safety strategy. Furthermore, measuring the effectiveness of safety management will require to assess how the different elements of the extended safety fractal are aligned to implement that strategy, in order to fit the specific mission and sector [55]. A similar focus on the need for strategic alignment can be found in the integrated performance management framework [40] presented by Verweire as a prerequisite for excellent performance. Depending on the situation and the scope and context of the assessment, the focus may move from the whole system towards more detailed implementation and control processes of the previously identified (extended) safety fractal, and vice versa, making optimal use of the self-similar attributes of the proposed models.

With SMS still the cornerstone of regulatory safety management obligations in several high-risk industries, assessments using the developed models can help to move from pure compliance to a more integrated approach based on dialogue. As mentioned earlier, there is no point in regulating the informal aspects of safety management. However, the extended safety fractal, in particular, offers the opportunity for safety authorities to develop a more holistic approach. An initial experience of one of the authors to develop adapted supervision strategies using the extended safety fractal, together with some national safety authorities within the European railway sector, promises positive results in that regard. 

In high-risk industries, a lot of time and effort also goes into incident investigation to collect very little useful information vis-à-vis explicit safety-risk management [17]. The main reason for this is that the scope of these investigations is often limited to the immediate causes and decision-making close to the adverse event, insufficiently addressing essential elements of safety management. The authors have already used the concept of the safety fractal to develop an innovative accident and incident analysis method that guides investigators to explore the composite elements of an SMS in a natural and logic way [44,68]. Starting from the findings close to operations that explain the occurrence (being the elements accident investigators are first confronted with), the same simple five steps, inspired from the safety fractal, are iterated to evaluate the performance of relevant SMS processes. The idea of nested control loops (e.g., [69,70]) is then used to identify the relevant set of control and implementation processes that influenced the chain of events. This leads investigators through the different operational, tactical and strategical levels that together form a socio-technical system in a more intuitive way. Moreover, this allows the analysis of how actions and decisions taken by individuals or teams at all these levels are affected by their local goals, resource constraints and external influences [71] and to discover the “local rationality” of decision and policy makers. Haavik et al. [72] describes this as the need to combine in-depth studies of work with an understanding of the organizational settings in which this work takes place. This is expected to result in recommendations that address the capability of responsible organizations to manage safety critical variability, leading them towards more resilient performance. As such, application of the SAfety FRactal ANalysis (SAFRAN) method [44] promises to create a greater impact on improving global system safety by moving away from the traditionally identified countermeasures that protect a causal link with a barrier [28], hereby fully embracing the idea that safety is an emergent property.

## 5. Conclusions

Resilient performance cannot be managed or controlled directly, but can be managed indirectly through the known characteristics that lead to it. By comparing the model behind SMS with specific requirements for process capability, this paper identifies a safety fractal that can help to understand how to build resilience into the SMS at all levels of a socio-technical system. This requires that resilience is explicitly identified as the safety strategy to follow, which should then be consequently implemented through both formal and informal cultural enablers. This mechanism, as modeled through the extended safety fractal, offers a systematic and more comprehensive framework to understand, organize and assess safety and resilience performance management. Further research should identify how this could best be realized in a practical and cost-effective way.

Throughout, this design research has iterated the consecutive steps of: (1) targeted review of state-of-the-art literature, (2) (improved) design of the models and (3) validation through practical application and comparison with actual safety management and accident investigation practices. This allowed the identification of user needs together with testing of the usability of the concepts. The first, practical examples presented above, already demonstrate the potential of the developed models to (re)vitalize the concept of SMS and to make the most of its full power. Further research is ongoing in order to continue the validation of these models in an operational context.

So far, the developed safety fractal has mainly been applied through the SAFRAN accident analysis method. The authors believe, however, that the (extended) safety fractal offers the possibility to also assess an organization’s capability for resilience in earlier stages, e.g., by using scenario-based assessments. The main argument for this is the fractal nature of the innovative approach that allows the same basic components to be used for both system wide and more detailed analysis. In addition here, further research could lead to identifying the most effective way of doing this.

## Figures and Tables

**Figure 1 ijerph-17-05478-f001:**
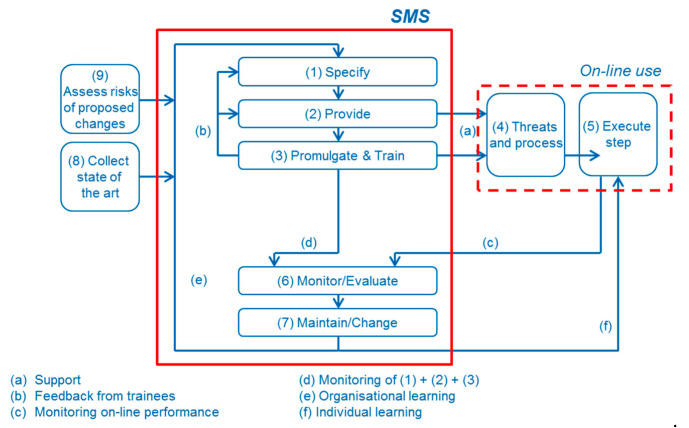
General structure of the delivery systems in an SMS (Safety Management System) [23].

**Figure 2 ijerph-17-05478-f002:**
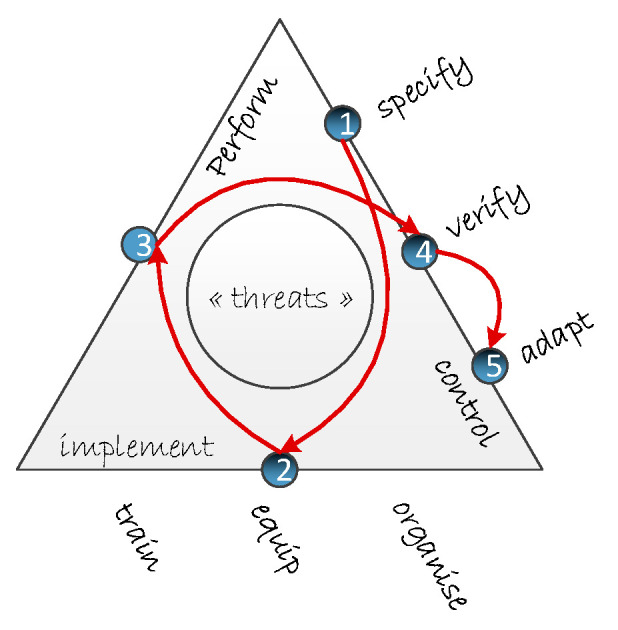
The Safety Fractal. Reprinted from the Safety Science 119, Accou, B.; Reniers, G., Developing a method to improve safety management systems based on accident investigations: The Safety Fractal Analysis, 185–293., Copyright (2020), with permission from Elsevier [44].

**Figure 3 ijerph-17-05478-f003:**
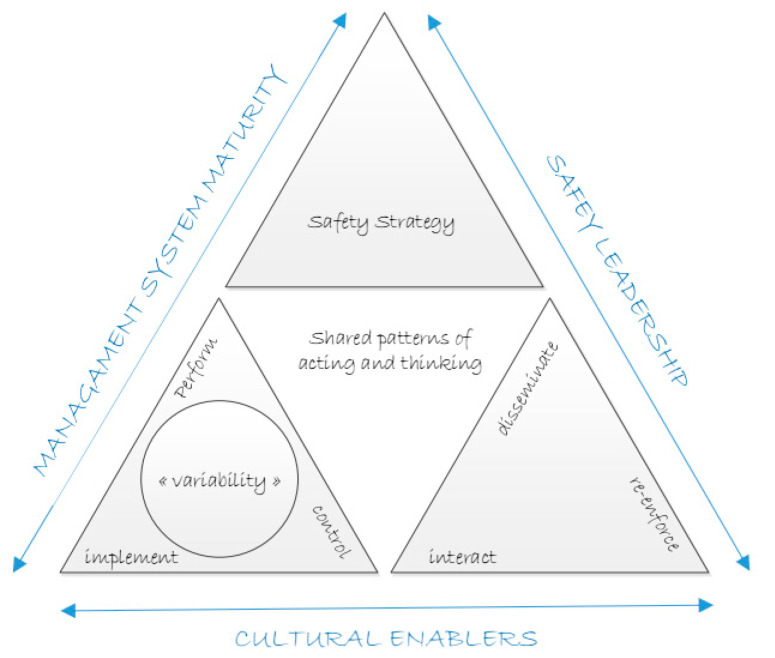
The Extended Safety Fractal.

**Table 1 ijerph-17-05478-t001:** ISO 15504 capability levels and attributes [32].

Level 0: Incomplete process The process is not implemented, or fails to achieve its process purpose.	
Level 1: Performed process The implemented process achieves its process purpose.	PA 1.1 Process performance attribute
Level 2: Managed process The previously described Performed process is now implemented in a managed fashion (planned, monitored and adjusted) and its work products are appropriately established, controlled and maintained.	PA 2.1 Performance management attribute PA 2.2 Work product management attribute
Level 3: Established process The previously described Managed process is now implemented using a defined process that is capable of achieving its process outcomes.	PA 3.1 Process definition attribute PA 3.2 Process deployment attribute
Level 4: Predictable process The previously described Established process now operates within defined limits to achieve its process outcomes.	PA 4.1 Process measurement attribute PA 4.2 Process control attribute
Level 5: Optimizing process The previously described Predictable process is continuously improved to meet relevant current and projected business goals	PA 5.1 Process innovation attribute PA 5.2 Process optimization attribute

PA—Done process attributes.

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
