# Peer review of "Introducing the Extended Safety Fractal: Reusing the Concept of Safety Management Systems to Organize Resilient Organizations"

_ijerph, 2020, doi:10.3390/ijerph17155478_

Round 1
Reviewer 1 Report
No doubt, authors have improved the manuscript and addressed almost all the observations, however, manuscript still requires to be proofread by an English language expert before possible publication.
Author Response
The revised version of the manuscript has been proofread by an English native speaker, with a master degree in Organisational Psychology and extensive knowledge on human factors and the implementation of SMS.
Reviewer 2 Report
The authors introduced proper changes to the paper. In my opinion, the paper after rewriting, presents the good academic level and may be published. Some editing comments below:
I recommend to write ‘First Age, Second Age …’ – starting with big letters like in the name.
I suggest to write words originating from ‘behaviour’ in British way – with ‘u’, unless originally (like references) written otherwise.
Author Response
The suggested changes have been made.
Reviewer 3 Report
This is an interesting and well redacted paper that propose a holistic framework for safety: the extended safety fractal.
Based on a previous study published recently in Safety Science (Accou, B., Reniers, G. Developing a method to improve safety management systems based on accident investigations: The SAfety FRactal ANalysis. Safety Science 2019, 115, 185-293.), the authors extend the Safety Fractal concept to try to organize resilient organizations.
Although I think this paper merits publication with minor changes detailed below, I think that the subject is not within the scope of International Journal of Environmental Research and Public Health. I would suggest to send the paper to Safety (within MDPI group) where it would have a larger audience.
Minor changes:
*Please, define PDCA the first time that it appears in the paper.
*Complete name in line 362.
Author Response
The proposed changes have been made.
This manuscript is a resubmission of an earlier submission. The following is a list of the peer review reports and author responses from that submission.
Round 1
Reviewer 1 Report
The paper has an interestic topic, not only of relevance for the research society but also in business daily practice. Occuptional Health and safety management is often considered as a third Wheel, which is dealt with apart from core activities. From that perspective, a simple method integrating safety management with the management of business development, based on scientifically delveloped theoretically methods and possible to use in practice, could have the potential to develop businesses at the same time as employees stay healthy, or at least avoid workrelated ill-health.
My major comment is related to how the paper is presented including numerous of theories related to the field. It is challenging to go through and understand the content of the paper, at the same time as you need to put the pieces together, to understand the arguement for why the safety fractional is suggested and how it is constructed. Are all theories relevant to include? Is it neccesary to present them in detail with all Component? Is it possible to reorganize the structure into fewer and more concrete sections? The theoretical part is very long and needs to be shortened or more concrete and specific to grasp.
The model is suggested for high risk industries. Is the model only applicable in that setting or would it also suit companies in other sectors? If not suitable in other sectors, why not? Several countries have safety management systems regulated by national laws. What would be the implications from a policy perspective?
The discussion includes new theoretical discussions based on theories not mentioned earlier in the manuscript. It also included an extended version of the Safety fractional not previously presented. The model, and relevant theories to argue for it, should preferrably be moved to the theory section and/or the result section.
Reviewer 2 Report
Please see the reviewer's comments in the attachment. Good Luck

Reviewer 3 Report
The authors of this paper start by exploring the concept of Safety Management System in a thorough and well documented way. Their point of view is well balanced; on the one hand they identify the shortcomings and inconvenients of SMS’s, on the other hand they nevertheless acknowledge the advantages.
The parallel they draw between SMS’s and performance management specifications is very interesting. The concept of safety fractal is new to me, so I was very interested to read about it. Perhaps the only weakness of this paper is that the definition and the origin of the concept are provided quite some time into the paper (lines 320-321).
This paper is essentially theoretical in nature; I look forward to reading about concrete applications of the Extended Safety Fractal.
Overall this paper is very well written and strongly documented. Probably the best paper I have had to review in a long while. My present research into safety culture in the commercial fishing industry made this reading even more timely.
Typos:
- Line 47: the comma after [11-13] is superfluous
- Line 64: “related to human performance” (missing word)
- Line 82: “This belief is shared” (spelling mistake)
- Line 79-80 (and line 105): “an SMS”; line 95 (and line 101): “a SMS”: please standardize
- Line 96: “concepts such as management” (missing word)
- Line 102: specify “International Civil Aviation Organization” in full
- Line 108: please place the name of the author (Grote) in front of “[5]”
- Line 133: specify “Functional Resonance Analysis Model” in full
- Line 149: please specify what ARAMIS stands for
- Line 214: “ISO/IEC 15504”; line 215: “ISO/IEC 155041”; is the “1” in the latter superfluous, or do the authors wish to refer to Part 1 (ISO/IEC 15504-1)?
- Line 235: superfluous space in front of “Dutch”
- Line 243: Pei-Hui Lin (first names unnecessary)
- Line 249: “Dutch” model (lower-case m)
- Line 262: “Dutch Safety Management Model” (if using the proper name of the model with upper-case initials, move the quotes to the end on the name)
- Line 296: “The Figure 2 below” (inverted words)
- Line 304: “activities are normally performed” (inverted words)
- Lines 375-376: the position of the numbers 1 to 6 is somewhat confusing
- Line 382: “already longer accepted” (superfluous letters)
- Line 398: “system, herewith” (comma)
- Line 450: “that, like as earlier identified” (change word)
- Line 453: “sustainable and safe performance” (missing letter)
- Line 500: “Several authors have reviewed existing” (missing letters)
- Line 500: superfluous space between “methods” and “and”
- Line 515: for the first time the plural is used: SMSs; in many instances before this could have been done
- Line 519: “Tis is in line” (missing word)
- Line 543: “inspired on from the” (change word)
Minor details that will probably be corrected during final edition:
- In some places there are unnecessary empty lines (295, 305, 370, 407, 416, 490)
- Some paragraphs (lines 378-398; lines 436-453; lines 536-557; lines 568-575) have a different line spacing from the rest; any particular reason for this?
Reviewer 4 Report
General overview:
The paper is prepared on a good academic level but it seems, in great parts, to be rather the review describing the state of the art, rather than the typical scientific article.
The title correctly reflects the content of the article. The introduction guides the reader through the main issues provided in the paper. Chapters 2 and 3 are rather the state of the art presentation than the research description. The main research is presented in the chapter 4 (Discussion). I think that citations in the chapter containing conclusions should not appear or be exceptional. In this section the authors should focus on presenting the results of their own work rather than referring to literature. In my opinion such a structure of the paper could be improved.
Detailed remarks:
Figure 2 was published earlier, so the proper reference in its title should be provided.
Chapter 3.2. And what about safety culture? – I don’t think that the question suits here. I suggest ‘safety culture issues’ or ‘towards safety culture issues’.
Editorial aspects:
Various line spacing and double spaces between words – corrections should be made.
Editing mistakes:
5 – 7 authors affiliation refers only to no. 1,
13 – ‘I’ seems redundant,
453, 603 – double dots,
608 – ‘Frontiers’ instead of ‘Fontiers’
Reviewer 5 Report
The draft introduces the background of SMS, but the summary of the SMS-related research is insufficient. What results have been obtained in each of the existing areas of research in SMS? What are the gaps in the existing research? How does this paper fill these gaps? What is the contribution of this draft? In the introduction, the author should elaborate on the above questions. In this draft, I saw that the author introduced many SMS models, such as the Dutch Safety Management Model, Structured Analysis and Design Technique, and Management system maturity models. However, it is difficult for readers to find out what the author's contribution is and what is the theoretical innovation of this draft. This manuscript should be more explicit about what theory it is based on. If figure 3 is the main result of the article, I don't think it is appropriate to put it in the discussion section. The format of the draft needs to be further refined, with some paragraphs having smaller line spacing, such as line82-98 and line 324-331. There is punctuation in the title. (line 4)